# A Novel Bulk-Optics Scheme for Quantum Walk with High Phase Stability

**Andrea Geraldi [1,\*], Luís Diego Bonavena [1], Carlo Liorni [2] 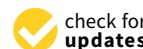, Paolo Mataloni [1] and Álvaro Cuevas [1]**

[1]   Physics Department, University of Rome La Sapienza, Piazzale Aldo Moro 5, 00185 Rome, Italy;
    ld.bonavena@gmail.com (L.D.B.); paolo.mataloni@uniroma1.it (P.M.); alvaro.phys@gmail.com (Á.C.)
[2]   Heinrich-Heine Universität, Institut für Theoretische Physik III, Universitätstraße 1, 40225 Düsseldorf,
    Germany; liorni@uni-duesseldorf.de
\*   Correspondence: andrea.geraldi@uniroma1.it

**Abstract:** A novel bulk optics scheme for quantum walks is presented. It consists of a one-dimensional lattice built on two concatenated displaced Sagnac interferometers that make it possible to reproduce all the possible trajectories of an optical quantum walk. Because of the closed loop configuration, the interferometric structure is intrinsically stable in phase. Moreover, the lattice structure is highly configurable, as any phase component perceived by the walker is accessible, and finally, all output modes can be measured at any step of the quantum walk evolution. We report here on the experimental implementation of ordered and disordered quantum walks.

**Keywords:** quantum optics; quantum walks; photonics

## 1. Introduction

The control of quantum systems allows not only exploring the interaction mechanisms of the microscopic world, but also exploiting the parallel coherent evolutions of such particularly sensitive systems, developing a new generation of information processing and communication [1,2], which will be capable of fast analysis [3] and dense data transport [4] with intrinsically-secure protocols [5]. For example, several quantum technology applications are currently under development, such as the implementation of search algorithms [6], the simulation of cellular automata [7] and the investigation of quantum diffusion processes, manifesting peculiar effects by the generation of topologically-protected quantum states [8,9].

These diffusion processes occur when the behavior of a quantum system is controlled by a surrounding environment, and it depends on its structure, on the interaction strength, and on the interference between all possible evolutions [10]. A well-known particular case is the so-called quantum walk (QW), representing the spatial random movement of a quantum particle in an n-dimensional lattice, where all degrees of freedom are commonly decoupled [11]. Depending on the lattice structure, a QW makes possible the simulation of important physical phenomena, such as Anderson localization [12,13] or a variety of other effects concerning different research fields [14].

In this work, we focus on the case of discrete QWs, where the quantum system only evolves at certain discrete times known as steps [15]. Here, we present a novel bulk optics scheme for simulations of QWs, exploiting unique features and presenting significant advantages with respect to current bulk [16] and integrated platforms [17].

## 2. Theoretical Model

The main elements of a QW are given by the walker, the coin, and the evolution operators of both the walker and coin [14,18]. In particular, for our one-dimensional case the walker corresponds to

the coherent superposition of all possible lattice sites occupied by the quantum particle during the evolution, and it is normally represented in the basis $\{|i\rangle_p\}$, spanning its associated Hilbert space $\mathcal{H}_p$. The coin corresponds to any degree of freedom enabling changes of position and is normally described by a two-level system with basis $\{|0\rangle_c, |1\rangle_c\}$ in the Hilbert space $\mathcal{H}_c$.

The QW evolution is controlled by the action of two operators: the first one acts on the coin state as a generalization of the Hadamard gate [19],

$$\hat{C} = \alpha \, |0\rangle_c \, \langle 0|_c + \beta \, |0\rangle_c \, \langle 1|_c + \gamma \, |1\rangle_c \, \langle 0|_c + \delta \, |1\rangle_c \, \langle 1|_c, \tag{1}$$

with $|\alpha|^2 + |\gamma|^2 = |\beta|^2 + |\delta|^2 = 1$ and $\alpha\beta^* + \gamma\delta^* = 0$; the second one corresponds to a shift in the walker position, conditioned to the state of the coin:

$$\hat{S} = |1\rangle_c \, \langle 0|_c \otimes \sum_i |i-1\rangle_p \, \langle i|_p + |0\rangle_c \, \langle 1|_c \otimes \sum_i |i+1\rangle_p \, \langle i|_p. \tag{2}$$

The operator $\hat{C}$, which can be considered as a physical device with two input and two output ports, can be optically achieved by encoding the coin basis in the path degree of freedom of a beam splitter (BS). Here, one can write $\alpha = \sqrt{R}e^{i\theta_\alpha}$, $\beta = \sqrt{1-R}e^{i\theta_\beta}$, $\gamma = \sqrt{1-R}e^{i\theta_\gamma}$, and $\delta = \sqrt{R}e^{i\theta_\delta}$, where $R$ is the BS reflectivity [20,21]. Then, by using $\theta_\alpha = \theta_0 + \pi/2$, $\theta_\beta = \theta_0$, $\theta_\gamma = \theta_1$, and $\theta_\delta = \theta_1 + \pi/2$ as solutions of the constraint $\theta_\alpha - \theta_\beta + \theta_\delta - \theta_\gamma = \pm\pi$, one can assume that all phase factors are effectively controlled only by $\theta_0$ and $\theta_1$, which could be prepared experimentally by external devices in the BS outputs $|0\rangle_c$ and $|1\rangle_c$, respectively. Since the absolute values of $\theta_0$ and $\theta_1$ are not accessible, only their difference $\theta = \theta_0 - \theta_1$ is relevant. Thus, a generic state $|\psi\rangle = \sum_{i,j} p_{i,j} |i\rangle_c \otimes |j\rangle_p$ with $\sum_{i,j} |p_{i,j}|^2 = 1$ evolves at each step (discrete time) according to:

$$|\psi\rangle' = \hat{U}_{c,p} |\psi\rangle, \tag{3}$$

where $\hat{U}_{c,p} = \hat{S} \cdot (\hat{C} \otimes \hat{\mathbb{I}}_p)$ represents the unitary evolution operator of an "ordered QW" [14,18].

In a more general scenario, each lattice coin can be different from the others and time dependent, as in the case of the completely-"disordered QW", where all lattice coins are random and independent at any step of the evolution [22]. Here, the state $|\psi\rangle$ evolves according to:

$$|\psi(t_{k+1})\rangle = \hat{U}_{c,p}(t_{k+1}, t_k) |\psi(t_k)\rangle, \tag{4}$$

with $\hat{U}_{c,p}(t_{k+1}, t_k) = \hat{S} \cdot \sum_p \hat{C}(t_{k+1}, t_k)_i \otimes |i\rangle_p \, \langle i|_p$, where $\hat{C}(t_{k+1}, t_k)_i$ is the coin operator at site $i$ for the step associated with the time interval $[t_k, t_{k+1}]$.

## 3. Experimental Implementation

The standard optical realization of a one-dimensional QW consists of a network of BSs, where each of them represents a particular position site of the lattice, with their ports encoding the coin basis and achieving the effective phase shift difference $\theta$ (see Figure 1). It is usually achieved by using photonics-integrated circuits, where a suitably-prepared array of microfabricated waveguides reproduces a fixed number of steps in a small volume [23], or by time-synchronization of photon pulses in bulk-fiber circuits [24], or in noisy large schemes of bulk optics only [25].

The actual setup implemented in this work consists of two displaced-multi-pass Sagnac interferometers (SIs) connected to each other through a common BS, as shown in Figure 2. Here, all beam trajectories are initially prepared in a collinear regime, and after a single mirror translation in the first SI ($SI_1$), one obtains the displaced loops lying in a single transmission plane, as used recently by us in another experiment [26]. This configuration is equivalent to a chain of Mach–Zehnder interferometers (MZIs), with intrinsically-stable phases that can be addressed independently in each mesh of the chain. The number of consecutive passages of light through the BS of Figure 2 determines unambiguously the length of the chain.

The one-dimensional QW of Figure 1 can be realized by the scheme of Figure 2, including as a further spatial dimension, the vertical direction, perpendicular to the horizontal plane of Figure 2. This is achieved by using suitably-designed beam displacers (BDs) intercepting some of the light trajectories in both SIs, namely clockwise trajectories in $SI_1$ and counter-clockwise ones in $SI_2$. For each passage of light through the BD, the number of QW sites grows by one unit, and the number of possible paths that the walker can go through is equal to $2^N$, where $N$ is the total number of steps. Thus, our scheme exploits the three dimensions of the same BS to increase the number of sites and steps of the QW. Individual phases can be easily addressed by using independently-rotating thin glass plates (RPs) in each QW mesh point (see Figure 2).

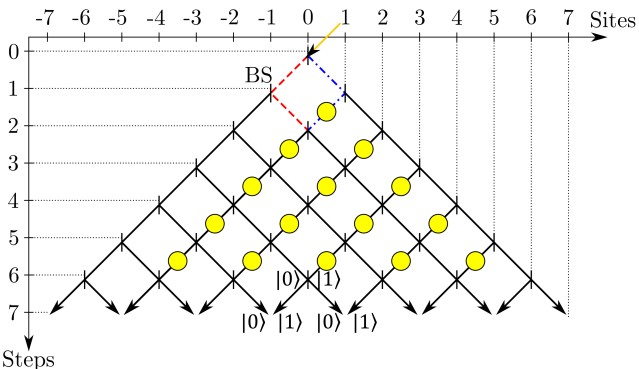

**Figure 1.** One-dimensional quantum walk (QW) in a beam splitter (BS) network: Arbitrary values of $\theta$ can be achieved by means of phase shifters, represented here by yellow circles. Notice that for each BS, the outputs $|0\rangle_c$ and $|1\rangle_c$ are automatically inverted in the following BS inputs as described by the operator $\hat{S}$.

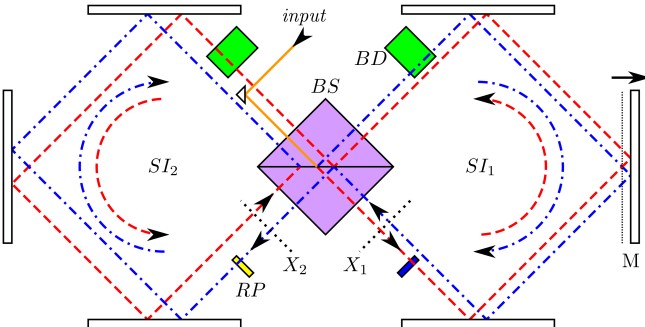

**Figure 2.** Double multi-pass Sagnac interferometer (SI): Both SIs are initially prepared in a condition of collinear alignment, then by a single translation of the mirror M in $SI_1$, one effectively obtains a scheme equivalent to a chain of Mach–Zehnder interferometers (MZI) exploiting the horizontal plane dimensions of the BS, as done in [26]. Odd step trajectories circulate in the first SI ($SI_1$), while the even ones circulate in the second SI ($SI_2$). The green rectangles represent the beam displacers (BDS), while the yellow one corresponds to the rotating glass plate RP that allows addressing the phases. The blue rectangle represents a fixed glass plate, mandatory to compensate time delay due to the thickness of the rotating plates (optical elements are not to scale). The red dashed and the blue dotted-dashed lines indicate the trajectories reproducing the first MZI of Figure 1.

Figure 3 shows the actual QW realized in the laboratory. In the system, BDs are realized by properly-oriented glass prisms. Additionally, the output radiation of each step can be extracted for measurement by a set of moving mirrors. In Figure 3a,b, different colors denote different transmission planes (light blue, green, violet, and red correspond to the beam in the ground, first, second, and third transmission plane, respectively). Two or more paths can exist in the same horizontal plane, but always

along different directions. The transverse spatial distribution of the QW internal paths arising in the setup is shown in Figure 4a,b, and it is obtained by cutting the paths in the $X_1$ ($X_2$) plane of $SI_1$ ($SI_2$) (see Figure 2) and looking at the BS. Here, the green boxes correspond to the sections of BDs intercepting only the clockwise (counter-clockwise) paths of the $SI_1$ ($SI_2$).

In the experiment, we have focused our attention on the case of single-photon evolution in ordered and disordered QWs. For this purpose, a proof of principle was performed using a CW laser at 810 nm as the input signal from a single-mode fiber coupler, as shown in Figure 3. An optical interference with visibility larger than 90% could be guaranteed up to the seventh step due to the good parallelism between the prism faces ($< 1\mu$ rad of deviation) and the very small time delay between the two arms of each MZI, which is orders of magnitude below the coherence length of the laser.

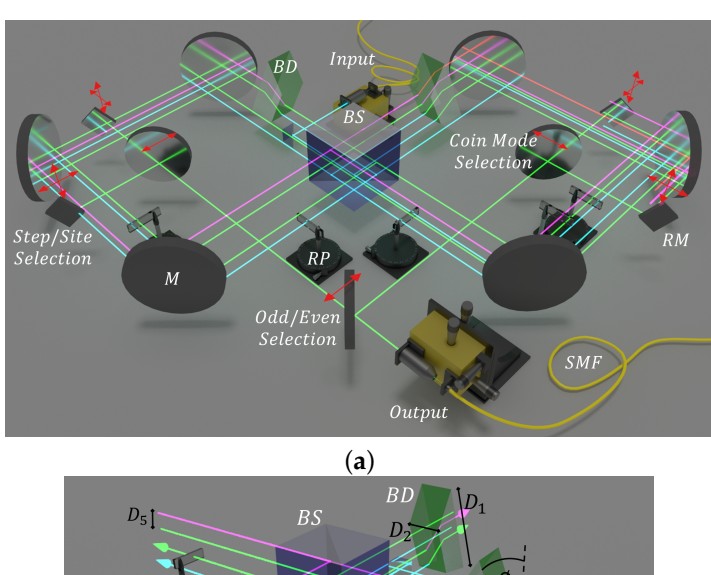

(**a**)

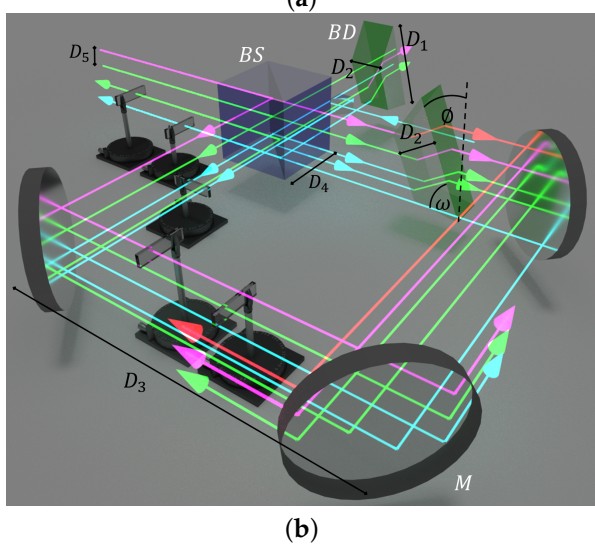

(**b**)

**Figure 3.** Complete scheme of the QW: (**a**) Photons enter the optical setup in the initial zero-plane. Later, only the clockwise (counter-clockwise) trajectories inside $SI_1$ ($SI_2$) go to an upper transmission plane by passing through an inclined dove prism of N-BK7 glass acting as *BD*, while the others pass through the rotating glass plates RPs with 1 mm of thickness. The modes of each step are extracted from the last column of beams circulating in each SI by a horizontal translation of the right angle mirrors (RM), while one can choose the position sites through their vertical translations. Site probabilities $P_i$ are extracted from the laser power after a single-mode fiber coupler. Here, light blue, green, violet, and red trajectories correspond to beams in the ground, first, second, and third transmission plane, respectively. (**b**) Relevant dimensions of the setup from the point of view of $SI_1$; $D_1 = 126.3$ mm, $D_2 = 30$ mm, $D_3 = 50$ cm, $D_4 = 5$ cm, $D_5 = 5.5$ mm, $\phi = 25$ deg, $\omega = 45$ deg.

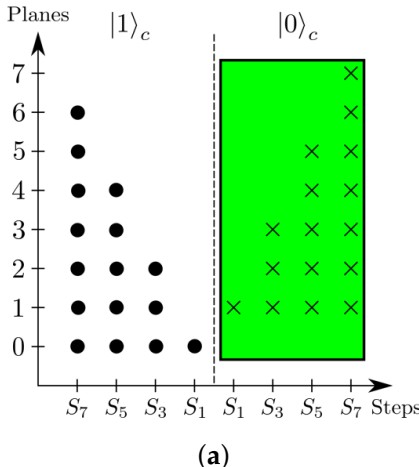
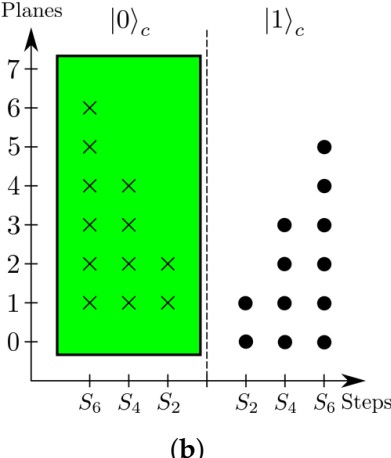

(a)　　　　　　　　　　　　　　　　　　(b)

**Figure 4.** Transverse spatial distribution of light beam trajectories in the apparatus: (**a**) Looking at the BS from the plane $X_1$ of $SI_1$. (**b**) Looking at the BS from the plane $X_2$ of $SI_2$ (see $X_1$ and $X_2$ in Figure 2). For each figure, the number of steps increases along the horizontal axis of the figure, going from inside to outside with respect to the central dashed line. Indeed, paths exiting from the BS at step $S_k$ are external with respect to the paths of the step $S_{k-2}$ (note that odd steps belong to $SI_1$, while even steps belong to $SI_2$). Columns with the same index $S_k$ show the modes of the $k^{th}$ step. Each step is represented twice because of the two possible states of the coin ($|0\rangle_c$, $|1\rangle_c$). Different planes of a column represent different sites, and the number associated with the site increases going from the bottom to the top. In this way, the same plane represents different sites for different steps. For example, in (**a**), the zero-plane for $S_1$ represents the site $-1$, but for $S_3$, the same plane represents the site $-3$. Point-marked trajectories ($|0\rangle_c$) go towards the viewer, while cross-marked ones ($|1\rangle_c$) go in the opposite direction. The green regions correspond to the effective transverse BD areas.

## 4. Results and Discussion

In Figure 5, we show the evolution of the walker state for the two cases of an ordered and a disordered QW, for an initial state $|\psi(t_0)\rangle = |1\rangle_c \otimes |0\rangle_p$. Here, the experimental data were obtained from the laser power measured on each site, where each of them has a contribution deriving from modes $|0\rangle_c$ and $|1\rangle_c$. The data corresponding to each time step $t_k$ have been obtained from the state $\rho_p(t_k) = Tr_c\left[|\psi(t_k)\rangle\langle\psi(t_k)|\right] = \sum_i P_i(t_k)|i\rangle_p\langle i|_p$ where the probabilities $P_i(t_k)$ for the occupation of site $i$ are normalized considering optical losses at each time $t_k$.

The results of Figure 5 show the typical distributions obtained for a ballistic and dispersive quantum transport corresponding to an ordered and disordered QW, respectively. They are in very good agreement with the theoretical predictions obtained by taking into account the actual parameters of the optical setup. We compared the experimental distributions with the theoretical ones for both ordered and disordered QWs through the similarity $S$, defined as $S = \left(\sum_{ij}\sqrt{G_i(s_j)G'_i(s_j)}\right)^2 / \left((\sum_{ij}G_i(s_j))(\sum_{ij}G'_i(s_j))\right)$, where index $i$ runs over positions and $s_j$ denotes different steps of the evolution. For the ordered case, where all phase factors were set to $\theta = 0$, we obtained a similarity value of $S = 0.990 \pm 0.002$. For the disordered case, where the phase factors alternate randomly their values between zero and $\pi$, the similarity was $S = 0.994 \pm 0.006$. The large values of $S$ demonstrates the very good agreement between experiment and theory.

In Figure 6, we report the variance of photon position during the evolution inside the lattice, expressed as $Var(t_k) = \sum_{i=-k}^{k} i^2 \cdot P_i(t_k) - \left(\sum_{i=-k}^{k} i \cdot P_i(t_k)\right)^2$. The graph allows us to compare the experimental data as a function of the number of steps with the theoretical predictions corresponding to a totally-symmetric QW and those obtained taking into account the real parameters of the setup. Data behave as expected from the theoretical simulation obtained with real parameters. A small

deviation is observed only for a large number of steps, probably given by the slight, but increasing spatial misalignment among all possible trajectories inside the setup.

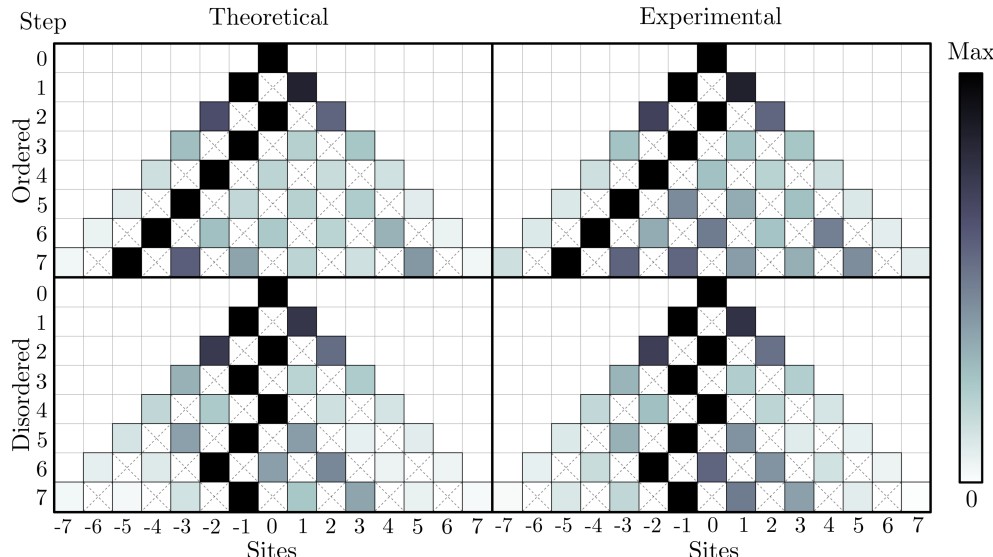

**Figure 5.** Walker distributions in the one-dimensional lattice: The QW starts as $|\psi(t_0)\rangle = |1\rangle_c \otimes |0\rangle_p$. The cross symbols denote the sites that cannot be reached by the walker at a certain step (even (odd) sites can be reached only at even (odd) steps). For better viewing, the values of probability at each step $t_k$ are normalized to the maximum value of probability obtained for the $t_k^{\text{th}}$ step. Both ordered and disordered QWs were extracted from a particular phase setting and by considering real parameters of the optical setup, in particular the BS reflectivity $R = 0.44$.

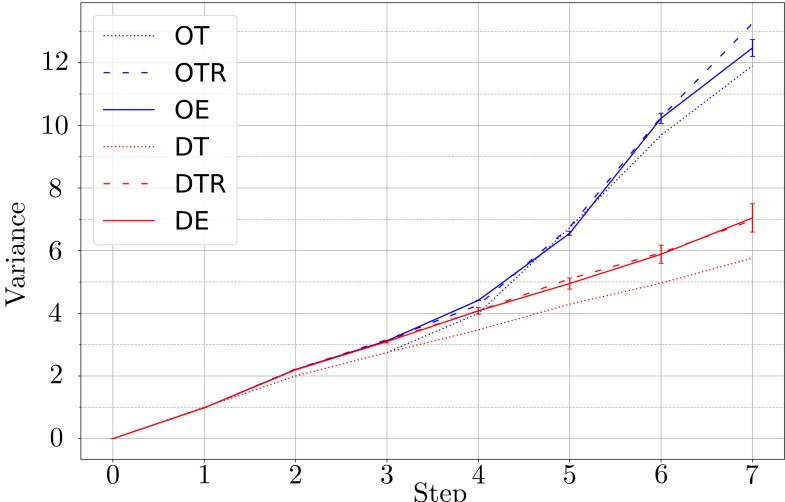

**Figure 6.** Position Variances: OT (DT): theoretical simulation for ordered (disordered) QW; OTR(DTR): theoretical simulation obtained by taking into account the parameters of the real optical setup for ordered (disordered) QW (here R stands for real); OE (DE): experimental data for ordered (disordered) QW. All the experimental points were obtained by averaging for each step the data obtained for each of three phase settings .

The relatively large size of the apparatus (see Figure 3b) was responsible of an observed small amount of phase instability. This effect was taken into account in the evaluation of the error bars of each experimental point.

In the case of ordered QW, experimental data were in good agreement also with theoretical predictions based on ideal optical elements, while a larger discrepancy was observed in the case of disordered QW. This was due to the non-symmetric BS used in the experiment, with measured reflectivity $R = 0.44$. The larger probability of photons to be transmitted at each passage through the BS introduced preferential directions followed by the walker. These trajectories were those characterized by a lower number of reflections, while the path with the largest probability was the one transmitted at each passage through the BS. The disordered QW was more sensitive to this effect if compared to the ordered one. Indeed, in the first case, the disorder tended to localize the walker around a single position, while the non-ideal BS brought the walker into a wide range of positions due to the preferential directions, thus increasing the variance. This resulted in a discrepancy between experimental data and predictions obtained with ideal parameters, which was strongly reduced once the non-ideal BS was introduced into the simulation. On the contrary, in the ordered case, preferential directions concurred with the ballistic behavior of the walker, resulting in a smaller discrepancy between experimental data and ideal theoretical predictions.

A different effect was expected in the case of a BS reflectivity larger than $0.5$. Indeed, an augmented reflectivity generated different preferential directions with respect to the $R < 0.5$ case where the path with the largest probability was the one reflected at each BS interface. In this way, the walker was confined around a small range of positions, resulting in a smaller variance with respect to the ideal case for both ordered and disordered QW. This is shown in Figure 7 where it is possible to compare the different variance behaviors expected for $R = 0.46, 0.5, 0.54$. As said, when $R < 0.5$, the variance was larger than the one calculated for the ideal case ($R = 0.5$), and the disordered QW was more affected by this effect if compared with the ordered one; when $R > 0.5$, the variance was smaller with respect to the ideal case for both ordered and disordered QW; however, in this case, the disordered QW was less affected by this effect when compared with the ordered one.

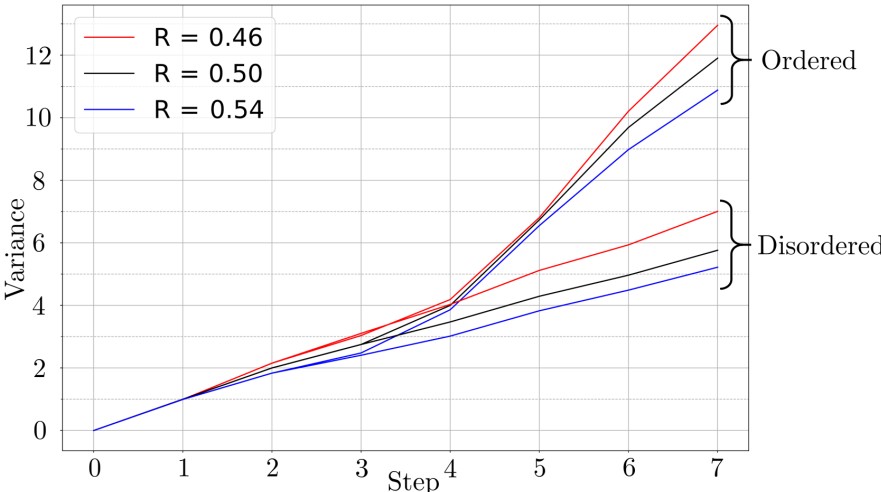

**Figure 7.** Expected variance behaviors corresponding to different values of R: the values of R are equal to $0.46, 0.5$ (ideal case) and $0.54$. The theoretical simulations for disordered QW are computed by a mean of 100 different phase settings.

## 5. Conclusions

In this work we have shown the feasibility of the bulk-optics implementation of a one-dimensional QW based on a novel double Sagnac interferometric configuration. We demonstrated its high level of customization, phase-stability and measurement accessibility, allowing at the same time to extract each coin mode at each position site and time step. This apparatus can manage more complex discrete QW scenarios, such as simulating two-dimensional lattices or studying two-particle interference by injecting the second photon in the other port of the BS. We estimated that by replacing the glass phase

control with suitably-designed spatial phase modulators, the total dimension of the system can be drastically reduced, arriving at up to 20 steps, which represents a valid alternative with respect to other platforms currently achieving more than 10 steps [27,28].

Henceforth, we expect to use this system to study exotic kinds of noise, to reproduce unusual kinds of quantum behaviors, like topologically-protected states [8], and to explore sub-diffusive and super-diffusive dynamics [29–32].

**Author Contributions:** A.G. coordinated the experimental implementation and achieved the data analysis. Á.C., P.M. and C.L. contributed principally to the concept and experimental design. L.D.B. and A.G. achieved the experimental measures. All authors contributed equally to the writing of the manuscript.

**Funding:** We acknowledge support from the European Commission Grant FP7-ICT-2011-9-600838 (QWAD-Quantum Waveguides Application and Development).

**Conflicts of Interest:** The authors declare no conflict of interest.

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
