# Peer review of "A Novel Bulk-Optics Scheme for Quantum Walk with High Phase Stability"

_condensedmatter, doi:10.3390/condmat4010014_

Round 1
Reviewer 1 Report
The authors experimentally demonstrated the discrete-time quantum walk with the displaced Sagnac inteferometers. This experimental approach seems to be really nice. I hope that more experimental results will be got from this experimental setup. As the experimental feasibility, how many steps can the authors implement? Also, the disordered or ordered discrete-time quantum walk was demonstrated. While this results are theoretically well known, the experimental demonstration is meaningful.
Finally, you remarked the super- or sub-diffusive behavior in quantum walk. Actually, according to the nonlinear quantum walk [Sci. Rep. 4, 4427 (2014)], the sub-diffusive behavior was seen. Therefore, the author guesses are partially right.
From the above reason, I can recommend that the slightly modified manuscript will be published.
Reviewer 2 Report
The authors present an experimental implementation of one-dimensional discrete-time quantum walk using an optical scheme based on coupled Sagnac interferometers. The topic of the paper is relevant, and the presentation is relatively clearly written. Therefore, I recommend the paper for publication upon the authors consider the optional comments listed below.
1. It was not quite clear to me at the beginning how exactly the set-up works. I believe the authors should clearly state want is the “vertical direction” (first sentence of the third paragraph of Section 3, and elsewhere) and what is the “horizontal plane” (caption of Figure 2, and elsewhere). As I understand, the horizontal plane is the plane of the paper, while the vertical direction is orthogonal to it, “pointing to the reader”. If so, then the beam displacers shift the beams along the vertical direction, thus separating the vertical spatial regions assigned to each step of the walk, effectively performing the shift operation. I believe the authors could describe the set-up in slightly more detail and rigour.
2. In connection to this, in the first sentence of Conclusions the authors write “…implementation of a 3-dimensional QW,…” As I understand, “3-dimensional” refers to the fact that the actual equipment (the experiment itself) “happens” in all 3 dimensions, while the QW achieved is one-dimensional.
3. It is somewhat difficult to follow what happens on Figure 3, as it is rather complex. But it seems to me that, at least at certain parts, the different horizontal beams (differing by their vertical position) have the same colour, thus “repeating” them without the apparent reason. In particular, I see two green beams at the BD right from the BS on Figure 3 a). Also, the (same) two green beams are visible on Figure 3 b), at the right side of the BS.
4. I do not understand the meaning of Figure 4. I believe the authors should explain it in more detail, both in the main text, as well as in the figure caption.
5. In the first paragraph of Section 4 the authors write that the probabilities are normalised. What exactly does it mean? That they all sum up to one? But the probabilities by definition sum up to one, unless other assumptions are added. What are the assumptions/causes that the probabilities do not sum to one, and have thus to be “normalised”? In connection to this, if G and G’ were (probability) distributions, then why writing the denominator in the expression for S on page 5? Should the two sums be manifestly equal to one?
6. In Conclusions the authors claim that “no other platform offers” (certain) characteristics that the proposed one does. Is this too strong a statement? With which platforms the current one was compared against, and what kind of comparative analysis was done?
7. I believe the normalisation factor 1/\sqrt{2} should be removed from equation (1). Also, below equation (2), $\theta_\gamma = \theta_0$ should be replaced by $\theta_\gamma = \theta_1$.
8. Throughout the manuscript, the authors predominantly use the term “quantum walk”, but sometimes the alternative “quantum random walk” is used. I believe the authors should unify the notation (preferably with the predominantly used form).
9. Finally, a few possible typos:
- I believe there is “with” missing between “connected” and “each” in the first sentence of the second paragraph of Section 3 (page 2).
- I believe that the second sentence of the last paragraph on page 3 should start with, say “For”, instead of “On”.
- I believe that the phrase “responsible of a small” from the third sentence on page 6 should be replaced by “responsible for a small”.
Round 2
Reviewer 2 Report
The authors have improved the manuscript and corrected minor typos. I recommend the paper for publication.
I have noticed only one possible typo in the penultimate sentence of the new caption for Figure 4. The point-like trajectories are denoted on the figure by |1\rangle_c, not by |0\rangle_c (and analogously for the cross-marked ones).